# MicroRNA-200c-5p Regulates Migration and Differentiation of Myoblasts via Targeting *Adamts5* in Skeletal Muscle Regeneration and Myogenesis

**DOI:** 10.3390/ijms24054995

**Published:** 2023-03-05

**Authors:** Yanwen Liu, Yilong Yao, Yongsheng Zhang, Chao Yan, Mingsha Yang, Zishuai Wang, Wangzhang Li, Fanqinyu Li, Wei Wang, Yalan Yang, Xinyun Li, Zhonglin Tang

**Affiliations:** 1Key Laboratory of Agricultural Animal Genetics, Breeding and Reproduction of Ministry of Education & Key Lab of Swine Genetics and Breeding of Ministry of Agriculture and Rural Affairs, Huazhong Agricultural University, Wuhan 430070, China; 2Shenzhen Branch, Guangdong Laboratory of Lingnan Modern Agriculture, Key Laboratory of Livestock and Poultry Multi-Omics of MARA, Agricultural Genomics Institute at Shenzhen, Chinese Academy of Agricultural Sciences, Shenzhen 518000, China; 3College of Animal Science and Technology, Shihezi University, Shihezi 832003, China; 4The Cooperative Innovation Center for Sustainable Pig Production, Huazhong Agricultural University, Wuhan 430070, China; 5Kunpeng Institute of Modern Agriculture at Foshan, Chinese Academy of Agricultural Sciences, Foshan 528226, China; 6Guangxi Engineering Centre for Resource Development of Bama Xiang Pig, Hechi 547500, China

**Keywords:** miR-200c-5p, *Adamts5*, skeletal muscle, regeneration, migration, differentiation

## Abstract

Skeletal muscle, as a regenerative organization, plays a vital role in physiological characteristics and homeostasis. However, the regulation mechanism of skeletal muscle regeneration is not entirely clear. miRNAs, as one of the regulatory factors, exert profound effects on regulating skeletal muscle regeneration and myogenesis. This study aimed to discover the regulatory function of important miRNA miR-200c-5p in skeletal muscle regeneration. In our study, miR-200c-5p increased at the early stage and peaked at first day during mouse skeletal muscle regeneration, which was also highly expressed in skeletal muscle of mouse tissue profile. Further, overexpression of miR-200c-5p promoted migration and inhibited differentiation of C2C12 myoblast, whereas inhibition of miR-200c-5p had the opposite effect. Bioinformatic analysis predicted that *Adamts5* has potential binding sites for miR-200c-5p at 3’UTR region. Dual-luciferase and RIP assays further proved that *Adamts5* is a target gene of miR-200c-5p. The expression patterns of miR-200c-5p and *Adamts5* were opposite during the skeletal muscle regeneration. Moreover, miR-200c-5p can rescue the effects of *Adamts5* in the C2C12 myoblast. In conclusion, miR-200c-5p might play a considerable function during skeletal muscle regeneration and myogenesis. These findings will provide a promising gene for promoting muscle health and candidate therapeutic target for skeletal muscle repair.

## 1. Introduction

Skeletal muscle, as the largest organ, occupied a profound role in the maintenance of physiological characteristics and homeostasis in mammals [1,2]. As is known, some factors, such as injury, stress, and exercise, can easily cause skeletal muscle damage [3]. After injury, skeletal muscle displays a remarkable regenerative capacity, which induces damaged parts to self-renew for repair [4,5]. However, it is incompletely clear for understanding the complex biological process of myogenesis and muscle regeneration. Therefore, exploring the regulatory factors and mechanisms of skeletal muscle regeneration is vital for the treatment of muscle health. 

So far, emerging studies have suggested that non-coding RNA affects skeletal muscle regeneration and myogenesis by regulating gene expression [6,7], such as microRNAs (miRNAs) [8,9], lncRNAs [10,11] and circRNAs [12,13,14]. Notably, miRNA is an important transcriptional or post-transcriptional regulator, exerting a crucial role in skeletal muscle regeneration and myogenesis. For example, down-regulated miR-204 enhances C2C12 myoblast proliferation and migration promoting skeletal muscle regeneration [15], and knockout of miR-223-3p results in impaired muscle regeneration [16]. As a new target, miR-155 affects the process of muscle regeneration by regulating the initial immune response [17]. Hence, miRNAs play profound effects concerning myocyte proliferation, migration, fusion, and differentiation in skeletal muscle regeneration. 

Our previous study has indicated that several miRNAs are differently expressed and involved in the regulation of skeletal muscle repair, especially miR-743a-5p [18]. In addition, miR-200c-5p is one of the important members belonging to the miR-200c family, the family closely associated with the proliferation, migration, and invasion of tumors and cancer cells. In particular, miR-200c-5p is considered as a new marker for the diagnosis of lupus nephritis [19]. In addition, miR-200c-5p has been identified as a prognostic indicator and therapeutic target for human hepatocellular carcinoma (HCC) patients [20]. Furthermore, researchers have demonstrated that miR-200c-5p inhibits the proliferation and metastasis of HCC via inhibiting the target gene MAD2L1 [21]. However, there are few reports on the roles of miR-200c-5p in skeletal muscle regeneration and myogenesis. 

In this study, we aim to elucidate the regulatory role of miR-200c-5p in skeletal muscle regeneration. The microarray results showed that miR-200c-5p is significantly down-regulated during C2C12 myoblast differentiation and it is highly expressed in skeletal muscle. In the process of skeletal muscle regeneration, miR-200c-5p expression first increased and subsequently decreased. Further, miR-200c-5p can promote myoblast migration and inhibit C2C12 myoblast differentiation in vitro. Mechanically, miR-200c-5p regulates the migration and differentiation of C2C12 myoblast via targeting the *Adamts5* gene in skeletal muscle regeneration and myogenesis. This study will develop a promising gene for promoting muscle health and candidate therapeutic target for skeletal muscle repair.

## 2. Results

### 2.1. Identifying miR-200c-5p as a Candidate Regulator for Skeletal Muscle Regeneration

In previous microarray data, we determined that 39 miRNA expression levels were significantly down-regulated during C2C12 differentiation (Figure 1A). Notably, some of the miRNAs have been reported to play important roles in skeletal myogenesis, such as miR-743-5p [18], miR-324-5p [22], and miR-499-3p [23]. Interestingly, GO analysis of Targetscan results showed that the predicted target genes of miR-200c-5p were involved in skeletal muscle regeneration and skeletal muscle fiber development (Appendix A). Moreover, miR-200c-5p is specifically highly expressed in the skeletal muscle of mice (Figure 1B) and highly conserved among different species (Appendix A). Based on these results, we constructed a mouse model of tibial anterior (TA) muscle injury to study the role of miR-200c-5p in skeletal muscle regeneration (Figure 1C). The H&E staining results showed that most muscle fibers were dissolved with the severe inflammatory response after the first to third day of injury. After the seventh day of injury, new muscle fibers were formed. After the 14th day the injury, muscle fibers were almost completely repaired (Figure 1D). In addition, *Pax7* as a marker gene for myosatellite cell activation was highly expressed at the early stage and the marker genes for myogenic differentiation (*MyoD*, *MyoG*, and *MyHC*) were highly expressed at the late stage during skeletal muscle regeneration (Figure 1E). Notably, miR-200c-5p displayed a similar or opposite expression pattern with these marker genes (Figure 1F). These results indicated that miR-200c-5p may be an important potential regulator for skeletal muscle regeneration and myogenesis.

### 2.2. miR-200c-5p Regulates the Migration of C2C12 Myoblast, but Does Not Affect Its Proliferation

The migration, proliferation, and differentiation of skeletal muscle satellite cells determine the ability of skeletal muscle regeneration [24]. The qPCR results showed that the miR-200c-5p mimics up-regulated the expression of myoblast migration marker genes (Figure 2A), while the expression level of these marker genes was down-regulated after miR-200c-5p was inhibited (Figure 2B). It indicates that miR-200c-5p regulated the expression of marker genes involved in migration. Consistently, the wound healing assay was performed to verify the role of miR-200c-5p in C2C12 myoblast migration. The results showed that the wound width of the miR-200c-5p mimic group was significantly shortened (Figure 2C,D), but the miR-200c-5p inhibitor group was significantly wider at 12 and 24 h after scratch treatment (Figure 2E,F). In addition, the transwell assay showed that the number of C2C12 myoblast passing through the membrane of the well increased significantly after transfecting with miR-200c-5p mimics after 12 h (Figure 2G,H). However, the number of C2C12 myoblasts was significantly decreased after miR-200c-5p was inhibited (Figure 2I,J). 

The mRNA and protein expression of marker genes Ki67 and Pcna involved in proliferation were not affected after miR-200c-5p was overexpressed (Appendix A) and inhibited (Appendix A) on C2C12 myoblast. Similarly, the CCK-8 assay also demonstrated no difference in the proliferation of C2C12 myoblast after transfecting with miR-200c-5p mimics and inhibitor at 24 h, 36 h, 48 h, and 72 h (Appendix A). These results suggested that miR-200c-5p regulates the migration, but does not affect the proliferation of C2C12 myoblast.

### 2.3. miR-200c-5p Modulates the Differentiation of C2C12 Myoblast

Next, we analyzed the effect of miR-200c-5p on C2C12 myoblast differentiation. We used horse serum to induce the C2C12 myoblast differentiation for 0, 3, 5, and 7 days (Figure 3A). During the differentiation of C2C12 myoblast, the expression of miR-200c-5p was gradually decreased, which displayed an opposite expression pattern with *MyHC* (Figure 3B). It is consistent with the results of the microarray. Based on these results, we speculated that miR-200c-5p might be related to the differentiation of C2C12 myoblast. To certify this hypothesis, miR-200c-5p mimics and inhibitor were transfected into C2C12 myoblast. The qPCR results showed that the expression of *MyoG* and *MyHC* were down-regulated under the overexpression of miR-200c-5p (Figure 3C), while knockdown of miR-200c-5p had the opposite effect (Figure 3D). Moreover, Western blot and immunofluorescence assays revealed that overexpression of miR-200c-5p inhibits MyHC protein expression (Figure 3E,F) and myotube formation (Figure 3I,J), while interference of miR-200c-5p on the contrary (Figure 3G,H,K,L). These results indicate that miR-200c-5p modulates the differentiation of C2C12 myoblast.

### 2.4. Adamts5 Is a Direct Target for miR-200c-5p

MiRNA functions by regulating the expression of target genes at the transcriptional and post-transcriptional level [25]. *Adamts5* and *Plxdc2* were identified as potential targets for miR-200c-5p based on microarray analysis and bioinformatic prediction using miRDB, TargetScan, and miRMAP programs (Figure 4A). It is noted that only *Adamts5* was highly expressed in the skeletal muscle of mice (Figure 4B), while Plxdc2 expression was low (Figure 4C). During the differentiation of C2C12 myoblast, *Adamts5* displayed an opposite expression pattern with miR-200c-5p. However, *Plxdc2* expression did not change significantly (Figure 4D). Subsequently, we transfected miR-200c-5p mimics and inhibitor into C2C12 myoblasts and detected the expression of the *Adamts5* and *Plxdc2* to further confirm the target gene. The qPCR results showed that *Adamts5* was down-regulated and up-regulated under overexpression and knockdown of miR-200c-5p (Figure 4E,F), respectively. However, the expression of Plxdc2 was not affected by miR-200c-5p (Figure 4E,F). In addition, the RIP assay also showed that *Adamts5* was significantly enriched in the miR-200c-5p overexpression group (Figure 4G,H), while *Plxdc2* was not enriched (Appendix A). Furthermore, the vectors containing the wild-type sequence (Adamts5-3’UTR-WT) or mutated seed sequence (Adamts5-3’UTR-MT) were constructed to perform dual-luciferase assay (Appendix A). The luciferase activity of the wild-type vector was markedly reduced under miR-200c-5p overexpression, but no changes in the mutation-type group (Figure 4I). Moreover, the protein expression of Adamts5 was consistent with its changes at the mRNA level (Figure 4J,L). Hence, these results suggested that *Adamts5* is a direct target for miR-200c-5p. 

### 2.5. Adamts5 Regulates the Migration of C2C12 Myoblast but Does Not Affect Its Proliferation

In the process of regeneration following skeletal muscle injury, we determined that the expression of *Adamts5* was first decreased and subsequently increased at both mRNA and protein level (Figure 5A,B). Next, we used functional gain and loss to study the effects of *Adamts5* on the proliferation and migration of C2C12 myoblast. The qPCR results showed that the expression of myoblast migration marker genes was down-regulated under the overexpression of *Adamts5* (Figure 5C). On the contrary, these marker genes were up-regulated after *Adamts5* was knocked down (Figure 5D). In addition, the wound healing experiment showed that *Adamts5* overexpression significantly slowed down the wound healing (Figure 5E,F), but knocking down *Adamts5* accelerated wound healing at 12 h and 24 h (Figure 5G,H). The transwell assay further demonstrated that *Adamts5* overexpression inhibited C2C12 myoblast passing through the membrane of well (Figure 5I,J), while *Adamts5* knockdown had the opposite effect (Figure 5K,L). 

In addition, the effects of *Adamts5* on the proliferation of C2C12 myoblast were investigated by qPCR, Western blot, and CCK-8 assays. We determined that both overexpression and knockdown of *Adamts5* did not affect the proliferation of C2C12 myoblast (Appendix A). Therefore, these results suggested that *Adamts5* regulates the migration of C2C12 myoblast, but does not affect its proliferation.

### 2.6. Adamts5 Is Involved in the Differentiation of C2C12 Myoblast

*Adamts5* expression was down-regulated in both microarray results and the process of C2C12 differentiation (Figure 4D). In the subsequent assays, the mRNA expression of *MyoG* and *MyHC* were up-regulated through *Adamts5* overexpression (Figure 6A) and down-regulated via *Adamts5* knockdown (Figure 6B). Moreover, Western blot and immunofluorescence assays also verified that the overexpression of *Adamts5* can promote the protein expression of MyHC and myotube formation (Figure 6C,D,G,H), while *Adamts5* interference had the opposite effect (Figure 6E,F,I,J). These results suggested *Adamts5* is involved in the differentiation of C2C12 myoblast which was consistent with the previous reports [26]. 

### 2.7. miR-200c-5p Regulates the Migration and Differentiation of C2C12 Myoblast via Targeting Adamts5

To further determine whether miR-200c-5p regulates C2C12 myoblast migration and differentiation through targeting *Adamts5*, we conducted the co-transfection assay of *Adamts5* overexpression vector with miR-200c-5p mimics and mimics-NC, respectively. The results showed that miR-200c-5p overexpression can rescue the effects of *Adamts5* on the myoblast migration marker gene expression at the mRNA level (Figure 7A). The wound healing assay demonstrated that overexpression of *Adamts5* slowed down wound healing, while this effect was reversed by miR-200c-5p mimics (Figure 7B,C). In addition, miR-200c-5p overexpression can counteract the promoting effect of *Adamts5* on *MyoG* and *MyHC* expression in C2C12 myoblast (Figure 7D–F). Overall, these results confirmed that miR-200c-5p regulates the migration and differentiation of C2C12 myoblast via targeting the *Adamts5* gene.

## 3. Discussion

In this study, we determined that miR-200c-5p regulates the proliferation, migration and differentiation of C2C12 myoblast by targeting *Adamts5* in skeletal muscle regeneration and myogenesis. Skeletal muscle regeneration in response to pathological stress and remodeling processes is critical for maintaining homeostasis [4,27]. As one of the ways of post-transcriptional regulation, miRNAs play an efficient regulatory role in skeletal muscle regeneration [8,28]. Here, we determined a critical novel candidate miRNA miR-200c-5p through miRNA microarray of C2C12 myoblast differentiation. Similarly, Hiroyuki Shibasaki et al. also illustrated miR-188 differently expressed in skeletal muscle cell differentiation concerning skeletal muscle regeneration by microarray analysis [29]. Then, we identified miR-200c-5p as specifically highly expressed in skeletal muscle through tissue profile. Importantly, several miRNAs have been reported to play an important regulatory role in skeletal muscle regeneration and myogenesis, which are also special and highly expressed in skeletal muscle, such as miR-1 [30], miR-133 [31] and miR-206 [32]. Functional miRNA expression is different during the process of skeletal muscle regeneration. For example, miR-26a and miR-24-3p are down-regulated from the first to third days after injury, and then increased accompanied by skeletal muscle repair [33,34]. In our results, miR-200c-5p rapidly responded to skeletal muscle regeneration and its expression reached to peak on the first day after injury. Thus, these results initially reflected that miR-200c-5p was a potential regulator involved in skeletal muscle regeneration and myogenesis.

Accumulating studies showed that miRNA can regulate C2C12 proliferation, migration, and differentiation, and then affect skeletal muscle regeneration. For example, miR-127 promotes the differentiation of C2C12 myoblast and myotube formation, enhancing skeletal muscle regeneration ability [35]. In addition, the miR-17-92 cluster affected mouse skeletal muscle regeneration by regulating C2C12 myoblast proliferation and myotube formation [36]. In our results, qPCR, CCK-8, wound healing, Western blot and transwell experiments showed that miR-200c-5p promoted migration and did not affect the proliferation of C2C12 myoblast, accordingly. Moreover, miR-200c-5p inhibited the differentiation of C2C12 myoblast, which was similar to the differentiation effects of miR-200c-3p in C2C12 myoblast [37] and buccal mucosal fibroblasts [38]. Thus, our results, combined with previous reports and studies, directly and indirectly indicated that miR-200c-5p plays a vital role in skeletal muscle regeneration. However, miR-200c-5p suppresses cell proliferation and migration in human HCC cells [21]. It is inconsistent with our results. Similarly, miR-200c-3p belonging to the miR-200c family inhibits the migration and invasion in retinoblastoma and renal carcinoma cells [39,40,41,42,43]. On the contrary, miR-200c-3p promotes endothelial cell migration in pancreatic cancer [44]. In addition, miR-378 promotes proliferation in skeletal muscle myoblast [45] while suppressing the proliferation, migration, and invasion of colon cancer cells [46]. Therefore, the biological function differences of miR-200c-5p may be attributed to various cell types. Above all, these results suggest that miR-200c-5p plays multiple functions in the skeletal muscle cells and is involved in the regeneration of skeletal muscle.

MiRNA functions by targeting mRNA 3’UTR of target genes [47,48]. We demonstrated that *Adamts5* is a target gene for miR-200c-5p by dual luciferase and RIP assays. Adamts5 is a new metalloproteinase belonging to the *ADAMTS* family [49], which is associated with tumor proliferation and migration. Abnormal expression of Adamts5 is an effective marker of lymphatic invasion and lymph node metastasis in colorectal cancer [50]. In lung cancer, the expression of *Adamts5* is up-regulated, which can promote the migration and invasion of tumor cells [51]. In human glioblastoma, *Adamts5* also promotes cancer [52,53]. However, our results showed that overexpression of *Adamts5* inhibited the migration of C2C12 myoblast and did not affect the proliferation of C2C12 myoblast. The function of *Adamts5* in the skeletal muscle cells is inconsistent with these reports. However the previous studies also reported that *Adamts5* plays an anticancer role by inhibiting migration, invasion, and angiogenesis of gastric cancer cells (GC) [54]. *Adamts5* can also inhibit HCC cell migration and blood vessel formations by down-regulating *VEGF* expression [55]. It suggests that *Adamts5* has various effects on the proliferation and migration of different cell types. As known, regeneration occurs due to proliferating myoblast migrating to their respective sites and aligning and merging to form multinucleated muscle fibers by membrane–membrane fusion [56]. Interestingly, *Adamts5* overexpression significantly increases the adhesion between GC cells and ECM, and inhibits GC cell migration and invasion [54]. Therefore, we inferred that *Adamts5* resulted in increasing adhesion between myoblasts and the stromal layer, and thus hindered myoblasts migration. In addition, microarray data showed *Adamts5* expression is up-regulated during C2C12 differentiation and subsequent experimental results also proved that *Adamts5* can regulate the differentiation of C2C12 myoblast which is consistent with the previous report. Moreover, the researchers proved that *Adamts5* promotes C2C12 myoblast differentiation by encoding versicanase, which hydrolyzes skeletal muscle extracellular matrix (ECM) and promotes contact between cell membranes [26]. This finding further implies that *Adamts5* inhibits C2C12 myoblast by increasing the adhesion between the myoblast and the stromal layer. However, this hypothesis needs to be verified in future studies. In addition, we also determined that overexpression of miR-200c-5p reversed the effects of *Adamts5* on migration and differentiation of C2C12 myoblast in rescue assay. Therefore, it is concluded that miR-200c-5p regulates the migration and differentiation of C2C12 myoblast by directly targeting *Adamts5* in skeletal muscle regeneration and myogenesis. However, we lack the results of skeletal muscle regeneration and myogenesis in miR-200c-5p and *Adamts5* knockout mice. In this way, the regulatory effects of miR-200c-5p and *Adamts5* on skeletal muscle can be further consolidated. We will complete this work in the future study.

## 4. Materials and Methods

### 4.1. Study Design

Our experiment type is animal experiment, and we conducted experiments related to animals. Eight-week-old C57BL male mice were purchased from the Guangdong Vital River Laboratory Animal Technology Co., Ltd (Guangzhou, China). and allowed to adapt to the environment for one week before experiments. The number of mice is 50. The grade of mouse is SPF. The specification of mouse is 25–30 g. All mice were housed at 26 °C constant temperature and 60% relative humidity with a 12 h light/12 h dark cycle and free access to food and water. 

All experimental animal procedures in this study were performed according to the guidelines of Good Laboratory Practice, and the animals were supplied with nutritional food and sufficient water. Animal feeding and tests were conducted based on the National Research Council Guide for the Care and Use of Laboratory Animals and approved by the Institutional Animal Care and Use Committee at Huazhong Agricultural University (protocol code SYXK(e)2020-0084).

### 4.2. Participants

We selected C2C12 myoblast and 293T cell from American Type Culture Collection (ATCC) and C57BL mice for in vitro and in vivo experiments. Firstly, we are devoted to the study of skeletal muscle. C2C12 is a common cell line in the laboratory and the preferred model for the proliferation and differentiation of myoblasts in vitro. It has the characteristics of rapid proliferation and strong myogenic differentiation ability. C2C12 is widely used in the study of myogenesis and skeletal muscle development, such as skeletal muscle regeneration [57], muscle fiber type transformation [58], and energy metabolism [59]. It indicated that the results from C2C12 are reliable and universal. The transfection efficiency of 293T cells is very high, and it is often used in dual luciferase experiments to study the targeting relationship between miRNA and target genes [60,61]. Secondly, mice are common experimental animals with low feeding costs, high genetic similarity with humans, and fast reproduction. Moreover, C57BL mice are inbred mice, and the gene similarity between different individuals is as high as 98.6%. Therefore, the differences between individuals are small, the experimental error is small, too, and the experimental results are more reliable. The results of C57BL were reliable. In addition, the mice reached sexual maturity at 6 to 7 weeks of age. The life health index of 8-week-old mice is great and all organs are mature, including skeletal muscle, which is suitable for the study of skeletal muscle development.

### 4.3. Study Variables

#### 4.3.1. Muscle Injury Model of Mice

The tibialis anterior muscle of C57BL male mice was injected with 20 μL of 10 μM cardiotoxin (MedChemExpress, New Jersey, USA) as previously described. The treated samples were collected at 0, 1, 3, 5, 7, and 14 days after injection under sterile condition.

#### 4.3.2. Hematoxylin-Eosin (H&E) Staining

The tibialis anterior (TA) muscle of mice was embedded with OCT and solidified at low temperature. The frozen sections were prepared into 10 μm and fixed with 4% paraformaldehyde (Beyotime, Beijing, China) for 30 min. Then, the sections were stained according to the instructions of the hematoxylin and eosin (H&E) staining kit (Solarbio, Beijing, China). Finally, the sections were observed and photographed under a microscope.

#### 4.3.3. Cell Culture and Transfection

C2C12 myoblast and 293T cells were purchased from American Type Culture Collection (ATCC). The growth medium was Dulbecco’s Modified Eagle’s Medium (DMEM, Gibco, California, USA) containing 10% fetal bovine serum (FBS, Gibco, California, USA) and 1% penicillin–streptomycin (PS, Thermo Scientific, Massachusetts, USA), and the differentiation medium was DMED medium containing 2% horse serum (Bioind, Shanghai, China) and 1% PS. Then, the cells were cultured in a 37 °C cell incubator with 5% oxygen and 95% carbon dioxide. The transfection reagent was Lipofectamine™ 3000 (Thermo Fisher Scientific, Massachusetts, USA), operated according to its instructions.

#### 4.3.4. Plasmid and Oligonucleotides

pcDNA3.1-Adamts5, GLO-Adamts5-WT, and GLO-Adamts5-MT were synthesized by Gene Create (Wuhan, China). The control plasmid pcDNA3.1 was obtained from our laboratory. Adamts5 siRNA, siRNA-NC, all miRNA mimics, mimic NC, miRNA inhibitor, and inhibitor NC were obtained from RiboBio (Guangzhou, China). All sequences are provided in Appendix A.

#### 4.3.5. RNA Extraction and Real-Time Quantitative PCR (qPCR)

Cells or tissues were lysed with Triol (Invitrogen, Shanghai, China), extracted with chloroform, and denatured with isopropyl alcohol to precipitate RNA, then washed with 75% and 100% ethanol, respectively, and finally dissolved with DEPC water. The RNA quality and quantity were checked by NanoDrop 2000 (Thermo Fisher Scientific, Massachusetts, USA). The qualified RNA can be used for further study.

According to the instructions, HiScript III 1st Strand cDNA Synthesis kit (+gDNA wiper) (R312-01, Vazyme, Nanjing, China) and miRNA 1st Strand cDNA Synthesis Kit (MR101-01, Vazyme, Nanjing, China) were used for cDNA reverse transcription synthesis of mRNA and miRNA, respectively. In addition, mRNA qPCR was performed using Fast ChamQ Universal SYBR qPCR Master Mix (Vazyme, Nanjing, China) in a total reaction volume of 20 μL, including 10 μL 2 × SYBR Master Mix, 0.4 μL PCR Forward Primer, 0.4 μL PCR Reverse Primer, 2 μL cDNA, and 7.2 μL Sterile enzyme-free water. The reaction conditions were 95 °C for 30 s, then 95 °C for 10 s, and 65 °C for 30 s for 40 cycles; the reference gene was Gapdh. For miRNA, qPCR was performed using miRNA Universal SYBR qPCR Master Mix (Vazyme, Nanjing, China); the reaction system was the same as that of mRNA qPCR. The reaction condition was 95 °C for 5 min, then 95 °C for 10 s, 65 °C for 30 s for 40 cycles; the reference gene was U6. The relative expression levels of mRNA and miRNA were analyzed by the 2-ΔΔCT method. The sequence information of primers (Sangon Biotech, Shanghai, China) used for reverse transcription and quantification is shown in Appendix A.

#### 4.3.6. Protein Extraction and Western Blot

Protein lysate was used to lysate cells or tissues. The lysate consisted of RIPA buffer (Thermo Scientific, MA: Massachusetts, USA), phosphorylase inhibitor (Roche 5892791001, Basel, Switzerland), protease inhibitor (Roche 04693132001, Basel, Switzerland), etc. The concentration of the obtained protein was measured by the BCA kit of Biyuntian. Sodium dodecyl sulfate (SDS, CWBIO, Beijing, China) was added and denaturated at 100 °C for 20 min. The 8% and 10% sodium dodecyl sulfate-polyacrylamide gel electrophoresis (SDS-PAGE) gels (EpiZyme, Shanghai, China) were selected, and the sample size was 10 µL or 20 µL. After electrophoresis, the prefabricated adhesives were transferred to 0.45 μm Hybridization Nitrocellulose Filter (NC) membrane (Merck, New Jersey, USA), which was sealed with 5% skim milk powder, and then the primary and secondary antibodies were incubated. Primary antibodies Adamts5 (1:1000, Abcam ab41037, Cambridge, UK), KI67 (1:1000, Bioss bs-23102R, Beijing, China), PCNA (1:1000, Abcam ab18197, Cambridge, UK), MyHC (developmental myosin 1:1000, DSHB MF20, Iowa, USA) and Gapdh (1:1000, Abcam ab9482, Cambridge, UK) were diluted by 1× Tween (TBST) buffer (EpiZyme, Shanghai, China). Secondary antibodies were derived from rabbits and mice. ImageJ software (NIH, Bethesda, Maryland, USA) was used to analyze the gray value of protein bands.

#### 4.3.7. Wound Healing Assay

C2C12 myoblast was seeded into 6-well plates 24 h before transfection. After transfection, when cell confluence reached 90%, the cells were scratched along a straight line with a sterile 200 µL pipette tip. Then, the wounds were observed at 0 h, 12 h, and 24 h, and photographed with a 40x microscope (Olympus, Tokyo, Japan).

#### 4.3.8. Trans well

C2C12 myoblast was replaced with 1% dual antibody medium without serum after transfection for 12 h, and then starved for 12 h. After digestion, 1 × 10^5^ cells were transferred to the upper chamber of Transwell (Corning, New York, USA), and 500 µL DMEM medium containing 10% fetal bovine serum and 1% PS was added to the lower chamber. The upper chamber medium was 100 µL DEME medium containing 1% PS without fetal bovine serum. Then, the transwell upper chamber was taken out 12 h later, cleaned with PBS, fixed with 4% paraformaldehyde, and stained with DAPI. Finally, cell migration was observed under a 100x microscope, and statistical analysis was performed using Image J software.

#### 4.3.9. Cell Counting Kit-8 Proliferation Assay

C2C12 myoblasts were seeded into 96-well plates and harvested at 0 h, 24 h, 36 h, 48 h, and 72 h after transfection, respectively. The proliferation of myoblasts was measured using the Cell Counting Kit-8 (CCK-8) (Beyotime C0038, Beijing, China). CCK-8 reagent and complete medium were added to the 96-well plate in a mixture of 1:9 and incubated at 37 °C for 40 h. Every sample’s optical density (OD) at 450 nm was measured with a microplate reader, and the growth curve was drawn.

#### 4.3.10. Immunofluorescence Assay

The differentiated myoblasts were fixed with 4% paraformaldehyde (Beyotime, Beijing, China) at room temperature for 30 min. The cells were penetrated with 0.5% Triton-100 (Sigma, Shanghai, China) for 30 min and blocked with 5% bovine serum albumin (BSA) (Biofroxx, Berlin, Germany) for 1 h at room temperature. The primary antibody (mouse anti-myosin, MF-20-S, DSHB (1:500)) was incubated for 2 h at room temperature. The second antibody (anti-mouse, FITC, Servicebio (1:500)) was incubated at room temperature and shielded from light for 1 h. Finally, the nuclei were stained with DAPI (Beyotime, Beijing, China) for 10 min and then observed and photographed with a fluorescence microscope.

#### 4.3.11. RNA Immunoprecipitation (RIP)

C2C12 myoblasts were seeded into a T75 cell culture flask and transfected with miR-200c-5p mimics, and harvested cells after 48 h. RIP experiment was conducted according to the guidance of the Magna RIP™RNA-Binding Protein Immunoprecipitation Kit (Millipore, Bedford, MA, USA), and miRNA was pulled with anti-Ago2 antibody (Abcam ab186733, Cambridge, UK). Enrichment multiples of miR-200c-5p, Adamts5, and Plxdc2 were detected by qPCR.

#### 4.3.12. Double Luciferase Assay

293T cells were seeded on a six-well plate and co-transfected miR-200c-5p and miR-200c-5p-NC with Glo-Adamts5-WT by using Lipofectamine™ 3000 (Thermo Fisher Scientific, Massachusetts, USA) at 80% cell convergence. Another group co-transfected miR-200c-5p and miR-200c-5p-NC with Glo-Adamts5-MT. After 48 h, the cells were analyzed using the Dual-Luciferase Reporter Assay System Kit (Promega, Wisconsin, USA) on a GloMaxTM 20/20 Luminometer (Promega, Wisconsin, USA).

#### 4.3.13. Online Prediction of Target Genes

The target genes of miR-200c-5p in mice were predicted by online miRDB (http://mirdb.org/, accessed on 21 September 2020), TargetScan (http://www.targetscan.org/vert_71/, accessed on 21 September 2020), and miRMAP (https://mirmap.ezlab.org/app/, accessed on 21 September 2020).

### 4.4. Statistical Analysis

The fusion index was determined as the percentage of myonuclear in myotubes (defined as cells with two or more nuclei) in comparison with the total number of nuclei in the field and analyzed using the ImageJ software (NIH, Bethesda, Maryland, USA).

Image J was used to analyze the gray values of the protein bands according to the “Mean = IntDen/Area (Mean: Mean gray value; IntDen: Integrated Density)” formula. The operation process of Image J is as follows: (1) Open file; (2) Image-transform-rotate; (3) Image-Type-8-bit; (4) Analyze-Gels-Select First Lane; (5) Analyze-Gels-Plot Lanes; (6) Measure: use the “Line tool” to close the opening of the peak, use the “Magic wand” to click the peak in turn to obtain the gray value of the protein bands. Image J was also used to count the number of cells migrating in transwell assay. Image Pro Plus software (Media Cybernetics, Texas, USA) was used to analyze the width of the wound in wound healing assay.

GraphPad Prism 5.0 software (GraphPad Software, La Jolla, California, USA) was applied to all statistical analyses. All experiments were repeated at least three times. *t*-test and ANOVA were used to assess statistical significance, and *p* < 0.05 was considered statistically significant. The results are presented as mean ± S.E.M. * *p* < 0.05, ** *p* < 0.01, *** *p* < 0.001, *p* ≥ 0.05: ns (Not significant).

## 5. Conclusions

In conclusion, we determined that miR-200c-5p up-regulates in the early stage of skeletal muscle regeneration in mice, promotes migration, and inhibits differentiation of C2C12 myoblast. Mechanically, we demonstrated that miR-200c-5p regulates regeneration by affecting migration and differentiation of C2C12 myoblasts via targeting *Adamts5*. Our findings provide new insights into the role of *ADAMTS* family genes in skeletal muscle regeneration and myogenesis. Further, our study develops a promising therapeutic target and candidate gene for the muscular disease and animal genetics and breeding.

## Figures and Tables

**Figure 1 ijms-24-04995-f001:**
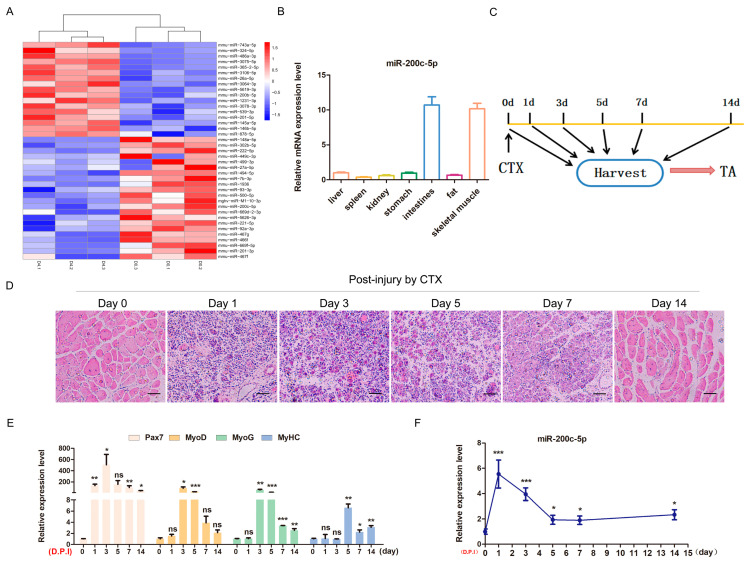
miR−200c−5p is a candidate regulator in skeletal muscle regeneration. (**A**) Cluster heat maps showing up− and down−regulated miRNAs on D4 of C2C12 cell differentiation and in undifferentiated cells. (**B**) Tissue expression profile of miR-200c-5p. (**C**) Schematic diagram of muscle injury experiments. CTX was injected into the tibial anterior (TA). (**D**) H&E staining results of TA muscle at different time points, magnification ×40. The gene expression profiles of marker genes (**E**) and miR-200c-5p (**F**) during skeletal muscle regeneration in mice. All experiments were repeated at least three times, glyceraldehyde-3-phosphate dehydrogenase (*Gapdh*) was used for normalization. The data are presented as mean ± S.E.M. * *p* < 0.05, ** *p* < 0.01, *** *p* < 0.001, *p* ≥ 0.05: ns (Not significant).

**Figure 2 ijms-24-04995-f002:**
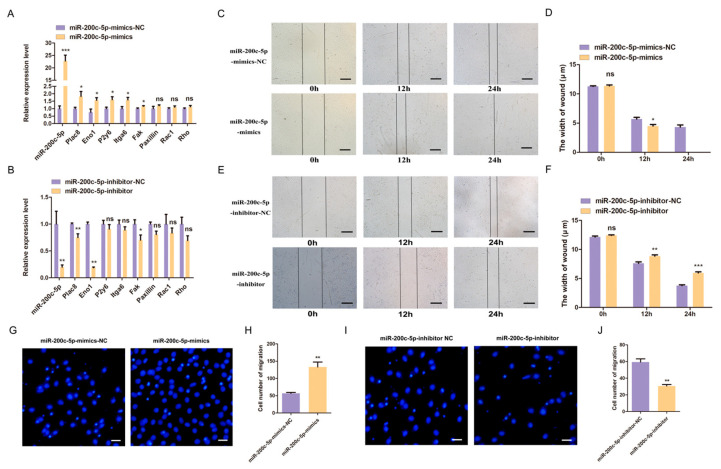
miR-200c-5p regulates the migration of C2C12 myoblast. The mRNA expression level of migration marker genes is enhanced by miR-200c-5p mimics (**A**) and inhibited by miR-200c-5p inhibitor (**B**). (**C**) miR-200c-5p overexpression influenced healing of the scratch at 12 h and 24 h in wound healing assay, magnification ×40. (**D**) Changes in C2C12 myoblast scratch width. (**E**) miR-200c-5p inhibitor influenced healing of the scratch at 12 h and 24 h in wound healing assay, magnification ×40. (**F**) Changes in C2C12 myoblast scratch width. (**G**) miR-200c-5p mimics influenced C2C12 myoblasts migration in transwell assays, magnification ×100. (**H**) The C2C12 myoblasts migration numbers. (**I**) miR-200c-5p inhibitor influenced C2C12 myoblast migration in transwell assays, magnification ×100. (**J**) The C2C12 myoblasts migration numbers. All experiments were repeated at least three times, Gapdh was used for normalization. The data are presented as mean ± S.E.M. * *p* < 0.05, ** *p* < 0.01, *** *p* < 0.001, *p* ≥ 0.05: ns (Not significant).

**Figure 3 ijms-24-04995-f003:**
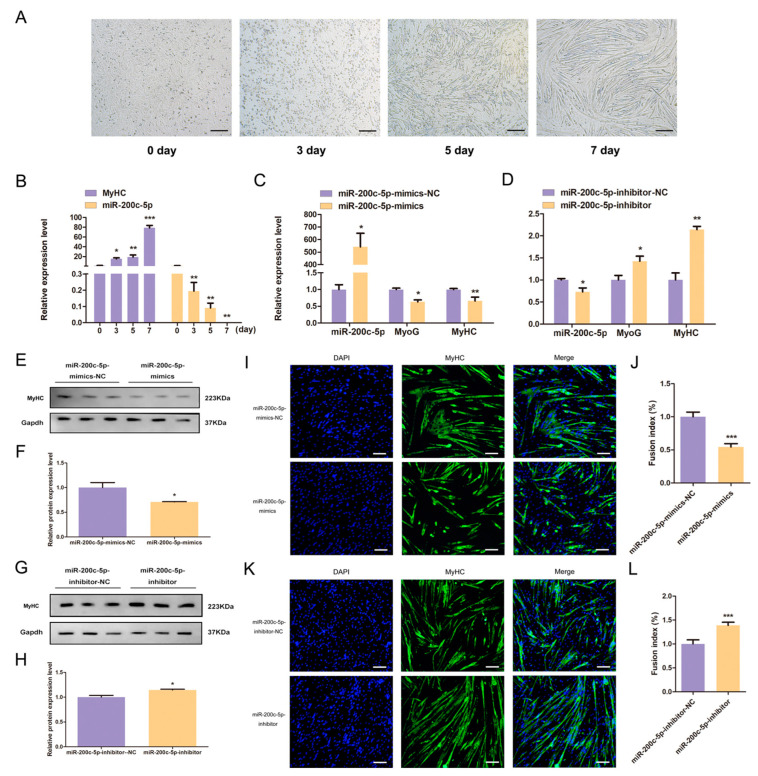
miR-200c-5p regulates the differentiation of C2C12 myoblast. (**A**) C2C12 myoblast differentiation was induced for 0, 3, 5 and 7 days by medium containing horse serum, magnification ×40. (**B**) The expression patterns of *MyHC* and miR-200c-5p in the process of C2C12 myoblast differentiation. The *MyoG* and *MyHC* mRNA expression were inhibited by miR-200c-5p mimics (**C**) and promoted by miR-200c-5p inhibitor (**D**). (**E**) The effect of miR-200c-5p mimics on MyHC protein expression. (**F**) The protein gray value was evaluated by Image J. (**G**) The effect of miR-200c-5p inhibitor on MyHC protein expression. (**H**) The protein gray value was evaluated by Image J. (**I**) Immunofluorescence results showed that miR-200c-5p mimics inhibited the myotubes formation. The nucleis of cell were stained with DAPI; magnification ×100 (Blue: DAPI; Green: MyHC). (**J**) Fusion index of C2C12 myoblast. (**K**) Immunofluorescence results showed that miR-200c-5p inhibitor promoted the myotubes formation. The nucleis of cell were stained with DAPI; magnification ×100 (Blue: DAPI; Green: MyHC). (**L**) Fusion index of C2C12 myoblast. All experiments were repeated at least three times, Gapdh was used for normalization. The data are presented as mean ± S.E.M. * *p* < 0.05, ** *p* < 0.01, *** *p* < 0.001, *p* ≥ 0.05: ns (Not significant).

**Figure 4 ijms-24-04995-f004:**
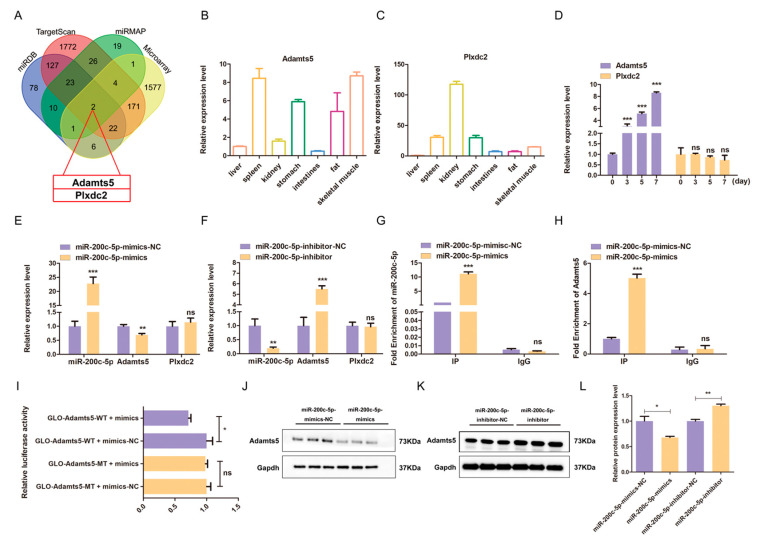
*Adamts5* is a direct target gene for miR-200c-5p in C2C12 myoblast. (**A**) The potential target genes of miR-200c-5p. (**B**) The tissue expression patterns of *Adamts5* and *Plxdc2* (**C**) in mouse. (**D**) The gene expression pattern of *Adamts5* and *Plxdc2* during C2C12 myoblast differentiation. The *Adamts5* gene expression, not *plxdc2*, was down-regulated by miR-200c-5p mimics (**E**) and up-regulated by the miR-200c-5p inhibitor (**F**). The enrichment multiples of miR-200c-5p (**G**) and *Adamts5* (**H**) in RIP experiment. (**I**) Dual Luciferase Reporter Assay to validate miR-200c-5p target at *Adamts5* 3’UTR in 293T. The effects of miR-200c-5p mimics (**J**) and inhibitor (**K**) on Adamts5 protein expression. (**L**) The protein gray value was evaluated by Image J. All experiments were repeated at least three times, Gapdh was used for normalization. The data are presented as mean ± S.E.M. * *p* < 0.05, ** *p* < 0.01, *** *p* < 0.001, *p* ≥ 0.05: ns (Not significant).

**Figure 5 ijms-24-04995-f005:**
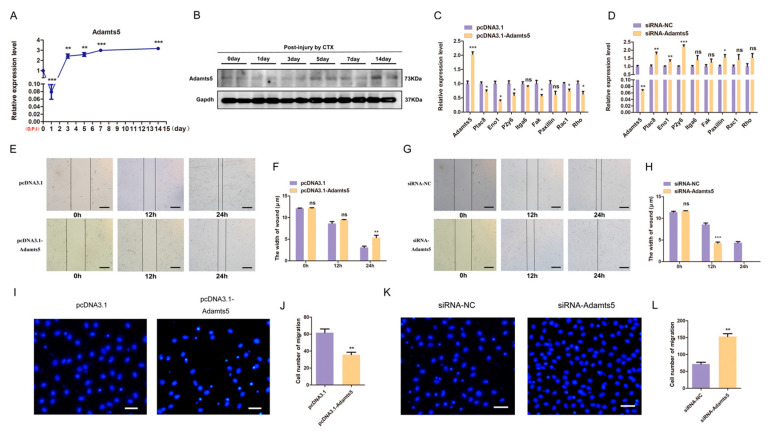
*Adamts5* regulates the migration of C2C12 myoblast. The expression profile of *Adamts5* in mouse skeletal muscle regeneration was reflected in mRNA (**A**) and protein levels (**B**), respectively. (**C**) The overexpression of *Adamts5* significantly down-regulated the expression of migration marker genes, while *Adamts5* knockdown on the contrary (**D**). (**E**) At 12 h and 24 h after scratch treatment, the effects of *Adamts5* overexpression on wound healing, magnification ×40. (**F**) Changes in C2C12 myoblast scratch width. (**G**) At 12 h and 24 h after scratch treatment, the effects of *Adamts5* knockdown on wound healing, magnification ×40. (**H**) Changes in C2C12 myoblast scratch width. (**I**) *Adamts5* overexpression reduced the number of C2C12 myoblasts migrating in transwell assays, magnification ×100. (**J**) The C2C12 myoblast migration numbers. (**K**) *Adamts5* interference increased the number of C2C12 myoblast migration in transwell assays, magnification ×100. (**L**) The C2C12 myoblast migration numbers. All experiments were repeated at least three times, Gapdh was used for normalization. The data are presented as mean ± S.E.M. * *p* < 0.05, ** *p* < 0.01, *** *p* < 0.001, *p* ≥ 0.05: ns (Not significant).

**Figure 6 ijms-24-04995-f006:**
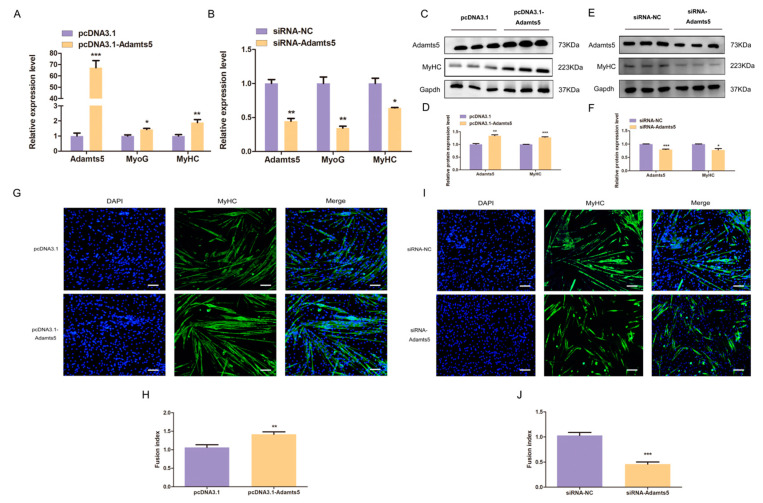
*Adamts5* is involved in the differentiation of C2C12 myoblast. The *MyoG* and *MyHC* mRNA expression levels were up-regulated by *Adamts5* overexpression (**A**) and down-regulated by *Adamts5* knockdown (**B**). (**C**) The effect of *Adamts5* overexpression on MyHC protein expression of C2C12 myoblast. (**D**) The protein gray value was evaluated by Image J. (**E**) The effect of *Adamts5* knockdown on MyHC protein expression of C2C12 myoblast. (**F**) The protein gray value was evaluated by Image J. (**G**) Immunofluorescence results showed that *Adamts5* overexpression promoted the formation of myotubes. The nucleis of cell were stained with DAPI; magnification ×100 (Blue: DAPI; Green: MyHC). (**H**) Fusion index of C2C12 myoblast. (**I**) Immunofluorescence results showed that *Adamts5* knockdown inhibited the formation of myotubes. The nucleis of cell were stained with DAPI; magnification ×100 (Blue: DAPI; Green: MyHC). (**J**) Fusion index of C2C12 myoblast. All experiments were repeated at least three times, Gapdh was used for normalization. The data are presented as mean ± S.E.M. * *p* < 0.05, ** *p* < 0.01, *** *p* < 0.001, *p* ≥ 0.05: ns (Not significant).

**Figure 7 ijms-24-04995-f007:**
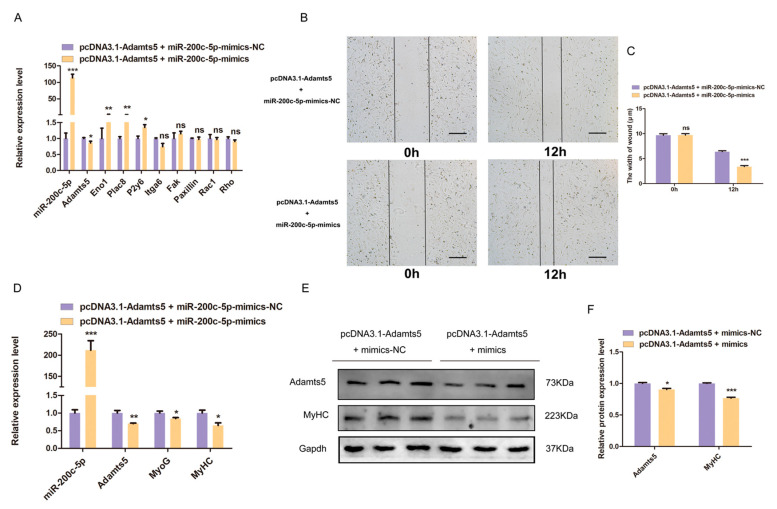
miR-200c-5p regulates the migration and differentiation of C2C12 myoblast via targeting *Adamts5*. pcDNA3.1-Adamts5 or pcDNA3.1 was cotransfected with miR-200c-5p mimics into C2C12 myoblast, and cultured with growth medium or differentiation medium. (**A**) The mRNA expression of C2C12 myoblast migration was examined by qPCR. (**B**) The migration ability was determined by wound healing assay, magnification ×40. (**C**) Changes in C2C12 myoblast scratch width. (**D**) The mRNA expressions of *MyoG* and *MyHC* were detected by qPCR. (**E**) Western blot showed the protein expression of MyHC. (**F**) The protein gray value was evaluated by Image J. All experiments were repeated at least three times, Gapdh was used for normalization. The data are presented as mean ± S.E.M. * *p* < 0.05, ** *p* < 0.01, *** *p* < 0.001, *p* ≥ 0.05: ns (Not significant).

## Data Availability

The data used to support the findings of this study are included within the article.

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
