# Peer review of "MicroRNA-200c-5p Regulates Migration and Differentiation of Myoblasts via Targeting Adamts5 in Skeletal Muscle Regeneration and Myogenesis"

_ijms, 2023, doi:10.3390/ijms24054995_

Round 1

Reviewer 1 Report

I would like to thank the authors for a very interesting article. The authors of this article have exhaustively searched the scientific literature, making a complete introduction that correctly puts the reader in the background on the topic developed. Also, in my opinion, the English language is correct, clear and understandable throughout the manuscript.

However, a series of modifications must be carried out to improve the quality of the study and for it to be accepted:

Point 4 "Materials and methods" should be point 2 of the work. There must be an order in the sections of a scientific article and this must be:

1. Introduction

2. Materials and methods

3. Results

4. Discussion

5. Conclusion

6. References

Similarly, the section "Materials and methods" should have some subsections:

2.1. Study design: specifying the type of study conducted and the reference number / code of the China Bio-377 logical Studies Animal Care and Use Committee.

2.2. Participants: explain the sample and the inclusion and exclusion criteria

23. study variables

2.4. Statistical analysis: the process carried out for the analysis of the data must be specified.

The results and discussion sections are complete with a wide exposition of figures that accompanies the results found. On the other hand, the discussion should begin with a paragraph specifying the objective of the study and the main findings and should end with the limitations.

Finally, the references section should be reviewed so that they are in accordance with the standards of the journal.

Reviewer 2 Report

The manuscript of “MicroRNA-200c-5p regulates migration and differentiation of myoblasts via targeting Adamts5 in skeletal muscle regeneration and myogenesis” by Ya. Liu and co-authors aims to elucidate the regulatory role of miR-200c-5p in skeletal muscle regeneration. The authors have significant background in the field of research of miRNAs, myogenesis, and muscle regeneration. In the present work, using miRNA microarray of C2C12 myoblast differentiation, the authors suggested that miR-200c-5p could be a novel post-transcriptional regulator involved in skeletal muscle regeneration and myogenesis. It was observed that overexpression of miR-200c-5p promoted migration, but it inhibited the differentiation of C2C12 myoblast. According to the results, miR-200c-5p rapidly responded to skeletal muscle regeneration and its expression reached to peak on the first day after experimental injury. The authors showed that miR-200c-5p was significantly downregulated in C2C12 myoblast differentiation and it was highly expressed in skeletal muscle. The authors discovered for the first time that that miR-200c-5p could regulate the differentiation and migration of C2C12 myoblasts via targeting the Adamts5 gene that encodes a new metalloproteinase belonging to the ADAMTS family. They found that the expression patterns of miR-200c-5p and Adamts5 were opposite in muscle regeneration, and the overexpression of Adamts5 inhibited the migration of C2C12 myoblasts, which may be due to increased adhesion between the myoblasts and the stromal layer.

The manuscript is interesting; all the conclusions are supported by the data obtained. The manuscript contributes to the systematization of modern knowledge about muscle diseases and provides a new understanding of the role of ADAMTS family genes in skeletal muscle regeneration and myogenesis.

Minor comments.

1.     Some statements contradict each other. Please, check this throughout the text. For example, Lines 71-72 vs. Lines 327-329: “Further, miR-200c-5p can promote myoblast differentiation and inhibit C2C12 myoblast migration in vitro. // In our results, qPCR, CCK-8, wound healing, western blot and transwell experiments showed that miR-200c-5p promoted migration and did not affect the proliferation of C2C12 myoblast, accordingly.”

2.     Lines 345, 347: In these cases, the Adamts5 enzyme is likely meant, but it is italicized (as a gene).

3.     The 4.16. Statistical analysis section could be improved.

Round 2

Reviewer 1 Report

The authors have responded to all my suggestions, but a series of minor modifications must be made in the "Materials and methods" section.

The first subsection should be called: 2.1. Study design and within it, the type of study and the information previously requested about the Ethics Committee must be entered.

The second subsection should be called: 2.2. Participants: in this section you must add the information that they have provided in the "author_response.pdf" document.

The third subsection should be called: 2.3. Study variables and within it add the sections from 2.2. up to 2.15 of the current manuscript

The last and fourth subsection must be that of Statistical analysis and the current information must be added.

Once this section is ordered, it will give a lot of quality to the manuscript.
